# Affordability of Food and Beverages in Mexico between 1994 and 2016

**DOI:** 10.3390/nu11010078

**Published:** 2019-01-03

**Authors:** M. Arantxa Colchero, Carlos M. Guerrero-López, Mariana Molina, Mishel Unar-Munguía

**Affiliations:** 1Center for Health Systems Research, National Institute of Public Health, Universidad No. 655. Col. Santa María Ahuacatitlán Cerrada Los Pinos y Caminera Cuernavaca, Cuernavaca 62100, Morelos, Mexico; cinys27@insp.mx (C.M.G.-L.); molinajaimes@hotmail.com (M.M.); 2Nutrition and Health Research Center, National Institute of Public Health, Cuernavaca 62100, Mexico; munar@insp.mx

**Keywords:** Mexico, affordability, food, beverages, energy density, income, nutrient-rich food index

## Abstract

We estimated the affordability of food and beverages by energy density and nutrient quality in Mexico and tested for differential trends in affordability over time and by income quintile. We used the National Income and Expenditures Survey from 1994 to 2016, and information on the caloric and nutrient content of food and beverages from Mexican food composition tables. We estimated food energy density (kcal/kg) and nutrient quality of food and beverages using the nutrient-rich food index (NRFI). Affordability of food and beverages was defined as household monthly income needed to purchase 1000 kcal. The affordability index was expressed by quintiles of energy density and quintiles of the NRFI and by income quintile over time. We found that more energy-dense foods and food and beverages with lower nutrient quality were more affordable compared to healthier food and beverages. Food categories with lower energy density and a higher NRFI became less affordable over time for most income groups, but the burden was higher for lower-income households. A comprehensive national strategy should be implemented to make healthier options more affordable and discretional foods and beverages with lower nutrient quality less affordable.

## 1. Introduction

Urbanization and economic development have driven major transformations in dietary habits and physical activity worldwide. Dietary patterns have shifted to higher consumption of refined carbohydrates, vegetable fats, added sugars, processed food, and meat, along with reductions in fiber intake from legumes, vegetables, and coarse grains [1,2]. These transformations in diets and physical activity have been associated with overweight and obesity, diabetes, and cancer globally [3,4].

In Mexico, along with a high prevalence of overweight and obesity affecting over 70% of adults and more than one third of children and adolescents [5,6], consumption patterns have shifted to a higher intake of ultra-processed food and beverages [7]. The consumption of sugar-sweetened beverages and nonessential energy-dense food accounts for 26% of total energy intake [8] and has been associated with a lower nutritional quality of diets in other countries [9].

One of the main drivers of food choice is price [1,10]. Several studies have shown that healthy diets cost more or are less affordable compared to lower-quality diets [11,12]. A recent systematic review found that in many countries, the cost per calorie of energy-dense food, which tends to be of poorer nutritional quality, is lower than that of nutrient-rich food [10]. In the United States, foods and beverages with higher quality based on nutrient content (higher in protein, fiber, and other micronutrients) cost more and are often consumed by high socioeconomic individuals compared to those with lower quality (higher in added sugar, saturated fat, and sodium), which are consumed more frequently among the poor [13,14]. A recent study for Mexico illustrated that less costly diets derive most calories from tortillas, "tamales", beans, and sugar, whereas higher-cost diets contain more calories from energy-dense food, processed foods, fruits and vegetables, and sugar-sweetened beverages [15].

To our knowledge, there are few studies from less developed countries on the affordability of food and beverages (the household income needed to purchase a specific number of calories from food and beverages) as well as how the affordability of food and beverages has changed over time and by income. A study conducted in 18 countries from all continents showed that fruits and vegetables are less affordable in low-income countries and in rural areas, but the paper did not show the affordability of other food and beverages and Mexico was not included [16].

The objective of this study was to estimate the affordability of food and beverages by energy density and nutrient quality in Mexico from 1994 to 2016. We also tested for differential trends in affordability over time and by income quintile.

## 2. Materials and Methods

### 2.1. Data Sources

We used thirteen rounds of the National Income and Expenditure Surveys (ENIGH for its acronym in Spanish) from 1994 to 2016 [17]. The ENIGH rounds have a two-stage stratified probabilistic design and are representative at the national level and by urban and rural strata. The surveys are collected every two years (except for a survey collected in 2005) between August and November, except for 1994, when data collection took place between September and December. Quantity and expenditures of household food and beverages purchased daily for consumption are collected for one week, including the monetary value of gifts, transfers, and the consumption of foods produced by the household. The unit of analysis is the household. Expenditures are reported by the household member responsible for purchasing food and beverages and complemented by individual members. Food and beverages in the ENIGH are classified into 222 categories composed of either a single item (for food and beverages widely consumed) or a group of items that are similar. We did not include expenditures on food and beverages purchased away from home, since there is no information on the quantity and type of food purchased in this category.

The energy content of foods was obtained from a dataset created by the National Council for the Evaluation of Social and Development Policies for the construction of the basic food basket in Mexico for 2009 [18]. The dataset contains average calories per 100 g (as purchased) for each food and beverage category of the ENIGH. Since the data compiled the energy content of food available in 2009 only, we used the same information for all rounds of the ENIGH.

The content of protein, fiber, vitamins A, C, and E, calcium, iron, potassium, magnesium, saturated fat, added sugar, and sodium of food and beverages was obtained from a food composition table compiled by the National Institute of Public Health [19]. The dataset was used to estimate the nutritional content of food reported in the National Health and Nutrition Survey in Mexico. This dataset compiled information from product packaging, food composition tables from the National Institute of Medical Sciences and Nutrition Salvador Zubirán, the United States Department of Agriculture (USDA) National Nutrient Database for Standard Reference, and the USDA Food and Nutrient Database for Dietary Studies. Since the data were estimated for 2012 only, we used the same information for nutrients for all rounds in the ENIGH.

We estimated energy density as the amount of energy measured in calories per kilogram for each food as purchased. Energy density was estimated only for solid foods; we excluded all caloric and noncaloric beverages from the estimation of energy-dense food, since water dilutes energy density values [20].

Since some high energy-dense foods may not be of low quality in terms of nutrient content—such as nuts—we also estimated the nutrient-rich food index (NRFI) developed by Drewnowski et al. [13]. The advantage of the NRFI is that it includes beverages that were excluded from the energy density analyses. The index estimates for each food and beverage category (per 100 calories) the sum of the percentage of its contribution to reference daily values for 9 recommended nutrients (protein, fiber, vitamins A, C, and E, calcium, iron, potassium, magnesium) minus the sum of the percentage of maximum recommended values for 3 nutrients that should be limited (saturated fat, added sugar, and sodium). We estimated the NRFI on 212 out of 220 food and beverages categories that were the same product as those listed in the food composition table compiled by the National Institute of Public Health.

### 2.2. Analyses

The affordability index was estimated as the ratio of the cost per 1000 kcal relative to monthly household income for each food and beverage category. Household income includes all sources of income received by members of the household as reported in the ENIGH.

To estimate the affordability index, we first calculated the cost of purchasing 1000 kcal for each food or beverage *i* purchased at the household level as reported in the ENIGH, expressed as calories per kilogram (kcal/kg) using the data set that contained calories. The cost per kg was approximated by deriving unit values (UV)—household expenditures for a specific food or beverage category *i* divided by the quantity purchased from the ENIGH. Unit values were aggregated at the municipality level to reduce potential household recall biases; therefore, the cost per 1000 kcal for each food and beverage was estimated at the municipality level. The cost per 1000 kcal is the result of dividing, for each food and beverage, the UV per kilogram by energy density (kcal/kg), multiplied by 1000, as defined in Equation (1),
(1)$${\mathrm{Cos}t}_{i}/1000\;\mathrm{kcal}=1000\times \frac{{\mathrm{UV}}_{i}/\mathrm{kg}}{{\mathrm{kcal}}_{i}/\mathrm{kg}}$$

The ENIGH collects information on household daily purchases of food and beverages for a week, but households may not purchase all food and beverages from the list of all categories in the survey. For example, some households may have purchased blackberries but others not. Because the affordability index measures how much income is needed to purchase 1000 kcal of a food or beverage category in the ENIGH, regardless of whether a household purchased it, we imputed within each municipality the energy costs of food and beverage categories not purchased by a household (blackberries, for example). If within a municipality, none of the households purchased a specific food or beverage, the energy cost would not be imputed, as energy costs were aggregated at the municipality level. The higher the affordability index is, the less affordable a food or beverage is.

Given the complexity of showing the affordability index for each of the 222 food and beverage categories, we expressed the index by quintiles of energy density (kcal/kg) and quintiles of the NRFI. We first estimated the energy density for each of the 222 food categories (except for beverages) from the sample of households and then created quintiles of energy density. The same method was used for the NFRI but including beverages. Since the sources of data used to estimate the NRFI and energy density were time-invariant, the quintile to which each food or beverage belongs did not vary over time. In descriptive analyses, we show the average affordability index by quintile of energy density and the same for the NFRI.

To test for differential trends in the overall affordability index over time and by income quintile, we ran a linear regression where the dependent variable was the affordability index for each food and beverage category at the household level. The independent variables were: binary variables for each round of the ENIGH survey, household income quintiles, interactions between income quintiles and this binary time variable to test if changes in the affordability index over time were different by income level. We adjusted the model for quintiles of energy density and of the NRFI. We controlled for household place of residence to account for differences in energy costs between urban and rural dwellers. We included education of the head of the household to adjust for the possibility that more educated individuals have greater abilities to find better products at lower prices, as described by Zhen [21]. We then derived the predicted values for the interactions between round and income quintile.

All estimations were weighted based on the ENIGH survey design. Monetary values were expressed as constant Mexican pesos (MXN) 2017 = 100, using the national consumer price index [22]. All analyses were conducted using Stata version 13 (Stata corporation, Texas, USA).

## 3. Results

The analytical sample between 1994 and 2016 were 247,164 households.

### 3.1. Affordability Index by Quintile of Energy Density and Quintile of the NRFI

Figure 1 shows the affordability index for food and beverages in Mexico from 1994 to 2016. Panel (a) shows the affordability index by quintile of energy density. We found that foods in the high and highest quintiles of energy density provides more affordable kcal: less income is needed to purchase kcal of foods with a higher energy density, such as animal based food, box cereals, sweet bread, snacks (see Appendix A with the list of foods by energy density quintile). By contrast, foods with the lowest energy density, such as vegetables and some fruits, are less affordable kcal. Foods with the lowest and low energy-dense food became less affordable over time (more income is needed to purchase 1000 kcal of food in 2016 than in 1994). Panel (b) shows the affordability index of food and beverages by NRFI quintiles. We found similar results: Foods and beverages with higher quality nutrient content are less affordable compared to those with a lower quality. This highest quintile of the NRFI includes mainly fruits and vegetables, whereas the lowest contains nonessential energy-dense food, sugar-sweetened beverages, processed meats, and sweet bread, among others (see Appendix A with the list of food and beverages by quintile of NRFI). The figure also shows that food and beverages of higher quality became less affordable over time.

In Appendix A, we show the cost per 1000 kcal (that was used to estimate the affordability index) by quintile of energy density and by the NRFI quintile. More energy-dense foods and of lower nutrient quality have lower energy costs.

### 3.2. Affordability of Food and Beverages over Time and by Income Quintile

Descriptive statistics of the sample in 2016 are presented in Appendix A. On average, 72.4% of the head of the household were male and 43.4% of them had completed primary school or less education and only 12.3% had a college degree or more. A total of 48.3% of household members were males and 17.1% were children under 12 years old. 78% of household lived in urban areas. Average household size in 2016 was 3.7 individuals.

The results from the regression model are presented in Table 1 and the predicted values for affordability from the interaction between income quintile and round are graphically displayed in Figure 2. For the lowest income quintiles, the affordability index increased over time: households had to spend, on average, 0.013 of their income in 1996 to buy 1000 kcal of food and beverages and 0.050 in 2016 (see coefficient for round). The interaction between round and income shows that the affordability index was lower (less income is needed to purchase 1000 kcal from foods and beverages) for all income quintiles compared to the lowest income quintile. The results also show that this difference became larger over time. For instance, in 1996, households in the highest quintile of income would have to spend 0.013 pesos less to purchase 1000 kcal of food and beverages and 0.054 pesos in 2016 than households in the lowest quintile of income. The affordability was lower for urban dwellers and household where the head was more educated. Compared to the lowest energy density quintile, the affordability index of foods was 0.07 pesos lower (more affordable) in the second quintile and the difference increased over quintiles III and IV and reached 0.10 pesos lower in the highest quintile of energy density. The affordability index of food and beverages was 0.013 pesos higher in the fifth quintile (less affordable) compared to the first quintile of the NRFI that has the lowest nutrient quality products. The affordability index was 0.008 pesos higher in the second quintile of the NRFI compared to the first quintile. However, food and beverages in the third and fourth quintile of the NRFI were less affordable compared to the first quintile.

## 4. Discussion

We estimated the affordability of foods and beverages by energy density and by the NRFI. We also tested for differences in overall affordability of food and beverages over time and by household income quintile. We found that more energy-dense foods and food and beverages with lower nutrient quality were more affordable, as less income is needed to purchase these food and beverages. Food categories with lower energy density and higher NRFI were less affordable for all income groups, but the poorest households had to spend significantly more of their income to purchase healthier products. Overall, food and beverages became less affordable over time, which can be explained by the increase in the affordability index of high nutritional quality and low density. The decrease in the affordability of food and beverages is seen for all income quintiles, but the difference over time was higher for the lowest income groups.

Between 2004 and 2016, food and beverages became less affordable, particularly among lower-income households (Figure 2). This concurs with the price liberalization of agricultural products in 2003, ten years after the implementation of the North American Free Trade Agreement, when subsidies that were intended to sustain market prices as well as subsidies based on production began to decline [23]. Price subsidies on fruit and vegetables are associated with an increase in the consumption of healthier food, but more studies are needed to determine its association with food affordability [24]. The reduction in the affordability of food and beverages for the lowest income is associated with the higher increase in food prices than inflation in Mexico [25] and not to the reduction in real income (based on ENIGH data, results not shown).

Based on the results of the cost per 1000 kcal (Appendix A), our findings suggest that for the same number of calories, people can get food and beverages of lower nutrient quality at lower energy costs. This may explain the high consumption of sugar-sweetened beverages and nonessential energy-dense food that represents 26% of total energy intake in Mexico [8]. The consumption of sugar-sweetened beverages has been associated with weight gain, diabetes, and other chronic diseases [26,27] and there is evidence of nondietary compensation: individuals do not reduce consumption in meals after drinking these beverages, which promotes overconsumption [28]. Similarly, diets high in energy density are associated with overweight and obesity, which could be explained by a passive overconsumption associated with an inability to reduce consumption on high energy-dense food [29,30,31].

We showed that nutrient-poor foods and beverages that provide the same number of calories are more affordable than foods higher in nutrient quality. The cost per calories or energy cost has been used as an indicator for comparing the cost of food and beverages in the last 100 years [11], although it could also be estimated per unit weight or serving. A study found that lower energy-dense food, such as fruits and vegetables, was more expensive than higher energy-dense food when the cost was estimated by calorie content but found the contrary when costs were assessed per serving (grams) [32]. However, when cost per unit weight is adjusted for both nutrient content and energy content, the results are consistent with our findings [11]. This does not imply that healthier options at low costs are not available. For instance, our data show that in the range of $18 to $22 Mexican pesos (about $0.95 to $1.16 USD), one can buy a kilogram of foods in the lowest quintile of energy density, such as spinaches, cucumbers, tomatoes, as well as foods in the highest quintiles of energy density, such as sweet cookies, sweet bread, and snacks. Although we are not assuming that people necessarily purchase food based on number of calories, the evidence shows that the consumption of energy-dense food and poor nutrient options is increasing [8]. Prices are key determinants of purchases [1,10], but there may be others, such as availability, income, marketing, palatability, satiety [33], and other costs (opportunity costs associated with time needed to purchase products, cooking, storage costs) that should be explored in future studies.

Our results are similar to other studies showing that high energy-dense food costs less, which represents a barrier to healthy eating [11,24,34,35,36,37,38]. A study conducted in 18 countries from all continents showed that fruits and vegetables are less affordable in low-income countries and in rural areas and that non-energy-dense food is less consumed because it is less affordable [16].

The study had some limitations. We acknowledge that, as in most household expenditure surveys, purchases of food and beverages in the ENIGH may be underreported. However, as methods for data collection in the ENIGH have not changed over time so the results and trends over time are comparable. We also recognize that we are missing all food and beverages purchased away from home that include meals consumed at restaurants, cafeterias, or any other food outlet, which increased from 14.6% in 1994 to 19.7% in 2016 [39]. Given that the ENIGH do not provide the specific food and beverages consumed away from home and given the heterogeneity of the items and the nutritional quality of the food offered in those outlets, we are unable to draw conclusions on how results would change were these purchases included.

Although the NRFI provides a score for each food and beverage based on the quality of the nutrients, results for some food and beverages deserve more discussion. For instance, juices and nectars with added sugars are classified in the fourth quintile of the NRFI because they have vitamins. Given the evidence that sugar-sweetened beverages are associated with weight gain, diabetes, and other chronic diseases [26,27], future studies should review the NRFI to evaluate how to score juices, nectars or other foods that combine recommended and non-recommended nutrients. We recognize that there are other nutrition profiling systems, such as the SAIN (score of nutritional adequacy of individual foods)-LIM (score based on nutrients to be limited) [40], in which nutrients are expressed 100 g as opposed to 100 kcal, which could be used in future studies to see how comparable the results are using different systems. However, these nutrition profiling systems have the same limitation of a low penalty on added sugar as the NRFI.

We decided to calculate quintiles of NRFI because it is very complex to show the affordability index for each score of the 222 food and beverage categories and for each round of the ENIGH. We recognize that one quintile could include foods of higher or lower quality, but quintiles differentiate well between low versus high quality. Additionally, Fulgoni [41] recommends that when using NRFI, it is helpful to separate scores by terciles, quartiles, or quintiles rather than using a score from large negative to large positive values.

Despite these limitations, to our knowledge, this is the first study on the costs and affordability of food and beverages in Mexico. We relied on a nationally representative survey with 13 rounds that guarantees comparability and consistency of the data over time.

If healthier food and beverages are and have become less affordable, particularly for lower income households, our findings suggest that poor families have been pushed to purchase more energy-dense food of lower nutritional quality. To revert this adverse scenario, a comprehensive national strategy should be implemented to make healthier options more affordable and discretional foods and sugar-sweetened beverages, which are of low nutrient quality, less affordable. Taxes on unhealthy food and beverages can be used to reduce purchases though increases in prices. The taxes on sugar-sweetened beverages and nonessential energy-dense food implemented in Mexico have shown reductions in consumption [42,43], but studies should evaluate the possibility of increasing these taxes, making sure they do not create socioeconomic inequalities, using revenues to compensate the poorest if needed. Moreover, revenues could be used to subsidize tap water (either to improve the quality or to provide the infrastructure for access in poor areas) as well as food whose consumption is below recommended guidelines in the country—such as fruits, vegetables, and legumes [8]. There is evidence that subsidies aimed at reducing the cost of fruits and vegetables could improve the quality of diets [44] and are associated with lower weight among children and adults [45]. However, more research is needed to design a food subsidy policy for healthy food that reaches the poorest population and assess its effect on consumption and health. In addition, interventions to provide information to consumers, such as a clear front of pack labeling, could improve consumers’ decisions towards purchasing healthier options [46,47].

## Figures and Tables

**Figure 1 nutrients-11-00078-f001:**
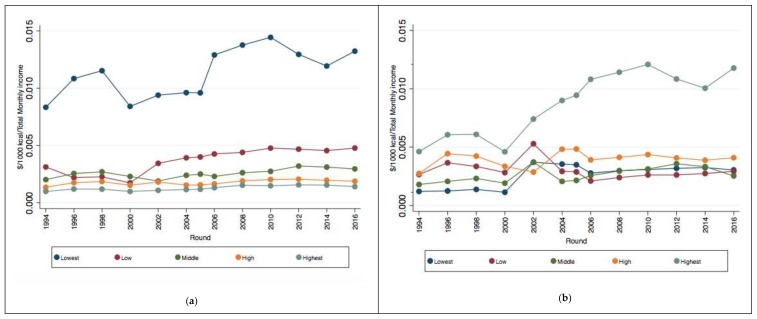
Affordability index ($/1000 kcal of food and beverages/household monthly income) of food and beverages, 1994–2016. Higher scores of the affordability index reflect lower affordability. (**a**) Affordability by quintile of energy density. (**b**) Affordability by quintile of the nutrient-rich food index. Authors’ estimations using the National Income and Expenditure Survey (1994–2016). Quintiles calculated on the 222 food categories.

**Figure 2 nutrients-11-00078-f002:**
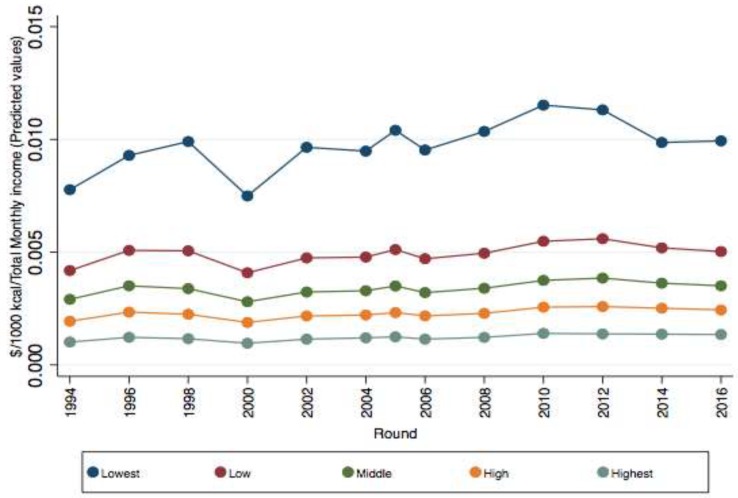
Predicted affordability index ($/1000 kcal of food and beverages/household monthly income) between 1994 and 2016 by income quintile, 1994–2016. Higher scores of the affordability index reflect lower affordability. Authors’ estimations using the National Income and Expenditure Survey (1994–2016). Predicted values for affordability from the interaction between income quintile and round (expressed as percent of income) using a linear regression adjusting for place of residence, education of the head of the household, quintile of energy density, and nutrient-rich food index (NRFI). All coefficients for the interaction between income quintile and round were significant at 1%. Estimations weighted based on the survey design.

**Table 1 nutrients-11-00078-t001:** Affordability ($/1000 kcal of overall food and beverages/household monthly income) by income quintile and round (1994–2016).

Variable	95% Confidence Interval			
	Coefficient	Lower Limit	Upper Limit	*p*-Value
**Round**				
1996	0.013	0.011	0.014	0.000
1998	0.018	0.017	0.020	0.000
2000	−0.003	−0.005	−0.002	0.000
2002	0.013	0.012	0.015	0.000
2004	0.007	0.006	0.009	0.000
2005	0.014	0.013	0.016	0.000
2006	0.042	0.040	0.044	0.000
2008	0.051	0.049	0.053	0.000
2010	0.065	0.063	0.067	0.000
2012	0.060	0.058	0.063	0.000
2014	0.041	0.039	0.042	0.000
2016	0.050	0.049	0.051	0.000
**Income quintile**				
II	−0.027	−0.028	−0.025	0.000
III	−0.037	−0.038	−0.036	0.000
IV	−0.044	−0.045	−0.043	0.000
V	−0.052	−0.053	−0.051	0.000
**Income quintile #round**				
II 1996	−0.008	−0.010	−0.007	0.000
II 1998	−0.015	−0.016	−0.013	0.000
II 2000	−0.001	−0.002	0.001	0.426
II 2002	−0.013	−0.015	−0.012	0.000
II 2004	−0.004	−0.006	−0.003	0.000
II 2005	−0.009	−0.010	−0.007	0.000
II 2006	−0.029	−0.031	−0.027	0.000
II 2008	−0.036	−0.038	−0.034	0.000
II 2010	−0.044	−0.046	−0.042	0.000
II 2012	−0.040	−0.042	−0.038	0.000
II 2014	−0.026	−0.027	−0.024	0.000
II 2016	−0.034	−0.035	−0.032	0.000
III 1996	−0.010	−0.012	−0.009	0.000
III 1998	−0.017	−0.019	−0.015	0.000
III 2000	0.000	−0.001	0.001	0.980
III 2002	−0.015	−0.017	−0.014	0.000
III 2004	−0.004	−0.006	−0.003	0.000
III 2005	−0.009	−0.011	−0.008	0.000
III 2006	−0.037	−0.039	−0.035	0.000
III 2008	−0.044	−0.046	−0.042	0.000
III 2010	−0.054	−0.056	−0.052	0.000
III 2012	−0.050	−0.052	−0.047	0.000
III 2014	−0.033	−0.034	−0.031	0.000
III 2016	−0.042	−0.043	−0.040	0.000
IV 1996	−0.012	−0.014	−0.011	0.000
IV 1998	−0.019	−0.020	−0.017	0.000
IV 2000	0.000	−0.001	0.002	0.732
IV 2002	−0.017	−0.018	−0.015	0.000
IV 2004	−0.005	−0.006	−0.003	0.000
IV 2005	−0.011	−0.012	−0.009	0.000
IV 2006	−0.042	−0.044	−0.040	0.000
IV 2008	−0.050	−0.052	−0.049	0.000
IV 2010	−0.062	−0.064	−0.059	0.000
IV 2012	−0.057	−0.059	−0.055	0.000
IV 2014	−0.039	−0.040	−0.037	0.000
IV 2016	−0.048	−0.049	−0.046	0.000
V 1996	−0.013	−0.015	−0.012	0.000
V 1998	−0.019	−0.021	−0.018	0.000
V 2000	0.000	−0.001	0.002	0.835
V 2002	−0.016	−0.018	−0.015	0.000
V 2004	−0.005	−0.006	−0.003	0.000
V 2005	−0.012	−0.013	−0.010	0.000
V 2006	−0.048	−0.050	−0.046	0.000
V 2008	−0.056	−0.058	−0.055	0.000
V 2010	−0.069	−0.071	−0.067	0.000
V 2012	−0.064	−0.067	−0.062	0.000
V 2014	−0.045	−0.047	−0.044	0.000
V 2016	−0.054	−0.056	−0.053	0.000
**Energy density quintile**				
II	−0.070	−0.070	−0.069	0.000
III	−0.079	−0.080	−0.079	0.000
IV	−0.092	−0.092	−0.091	0.000
V	−0.100	−0.100	−0.100	0.000
**NRFI quintile**				
II	0.008	0.008	0.008	0.000
III	−0.005	−0.006	−0.005	0.000
IV	−0.010	−0.011	−0.010	0.000
V	0.013	0.012	0.013	0.000
**Urban household**	−0.003	−0.003	−0.003	0.000
**Education of the head of the household**				
Secondary	−0.001	−0.001	−0.001	0.000
High school	0.000	0.000	0.000	0.846
College or more	−0.001	−0.001	−0.001	0.000
Constant	0.137	0.135	0.138	0.000

Authors estimations using the National Income and Expenditure Survey (1994–2016). Estimations weighted based on the survey design. Higher scores of the affordability index reflect lower affordability.

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
