# Peer review of "Affordability of Food and Beverages in Mexico between 1994 and 2016"

_nutrients, 2019, doi:10.3390/nu11010078_

Round 1
Reviewer 1 Report
Review of the manuscript
In Mexico, more energy dense foods and food and beverages with lower nutrient quality cost less compared to healthier options
Nutrients-385408
General comments
The topic is important but the manuscript is particularly difficult to follow because the methods aren’t correctly described.
Extensive rewriting is needed to end up with a paper describing a fully reproducible methodology. In particular, most of the analyses seem to have been done at the food level (based on food composition and price database), but others, especially those involving income and perhaps (but I am not sure) energy density seem to have been performed at the household level (based on household purchases, i.e. overall food basket). It is essential that this is clarified in each section of the manuscript, including the abstract
Title: remove the word “option” (in the title and elsewhere) because it suggests that you have been comparing healthy and less healthy versions of the same food (eg, full-fat dairy yogurt vs low-fat option, wholemeal bread vs white bread, etc..).
Specific comments
Lines 58-70: in the ENIGH, foods are reported as purchased. Is it also the case for foods in the food composition database of the National institute of Public Health? How did you manage the changes in weight (and therefore the changes in nutrient content) associated with preparation and waste (e.g. peeling, boning, water loss or gain during cooking, etc.)?
Lines 73-74: “the dataset contains average calories per 100 grams” => of food as consumed or as purchased?
Line 83-84: “We estimated energy density as the amount of energy measured in calories per kilogram” => of food as consumed or as purchased?
Line 84-86: unclear sentence (“We excluded all caloric and non-caloric beverages from the estimation of energy-dense food groups since the content of water is the food component that has a greater influence on energy density values.[19])
=> Line 85: what are the “food groups” doing here? What is the definition of a food group in this paper? It was not introduced before. You referred instead to 222 food categories (line 67).
=> do you rather mean that energy density was considered for solid foods only?
=> or, if it is the energy density of overall food purchases that wass calculated, do you mean that the energy density of food purchases was calculated based on solid food only???
Line 93: are the maximum recommended values for 3 nutrients that should be limited calculated per 100kcal or per 100g. For indicators of “bad” nutritional quality, it is know that it is not relevant to express them per 100kcal, as it gives bad scores for energy-dilute foods such as fruit and vegetables and lean meats… (see Drewnowski A, Maillot M, Darmon N. Should nutrient profiles be based on 100g, 100 kcal or serving size? Eur J Clin Nutr. 2009; 63(7):898-904.). If the score of nutrients to be limited was calculated based on 100kcal, I strongly recommend to test (as a sensitivity analysis) if the results are maintained with a score where the negative nutrients it is are expressed per 100g (e.g. as in the LIM score of the SAIN,LIM system, described in Am J Clin Nutr. 2009; 89:1227–36.)
Line 98: what is a “group”?
Line 105: why 10000 kcal and not 1000kcal, as in the equation?
Lines 108-110: Clarify that quintiles have not the same meaning in the two sentences: “The cost per 1,000 kcal and the affordability index were expressed by quintiles of energy density
(kcal/kg)”=> energy density calculated on the list of foods, on the list of 22 food categories, on the sample of household purchases?
“We also calculated the affordability indices by income quintile across the period”=> Quintiles calculated on what?
Line 109: the interest of calculating quintiles of NRFI is debatable. One would rather know which food have a good nutritional quality in an absolute sense and not which foods have a better nutritional quality. Hence, applying a relative definition of what is healthy is not useful (you may have an overwhelming place occupied by food of very bad nutritional quality in your food database, so that foods in the first quintile are not so good… ). Again, , I strongly recommend to test (as a sensitivity analysis) if the results are maintained with a nutrient profiling system classifying food in an absolute way (e.g. the SAIN,LIM system, described in Am J Clin Nutr. 2009; 89:1227–36.)
Lines 112-118: Not easy to understand how affordability by food group (or food category?) can be explained by household variables. Could you explain the rationale first.
Lines 123-125: to be removed
Line 127: the cost per 1,000 kcal was named the energy cost or the cost of dietary energy. It is important to use this terminology throughout the paper, and not simplifying by just “cost” (for instance at the end of line 128, but in many many other places in the manuscript), because cost refers to expenditure, which is really different.
Lines 129-130: the sentence “The higher costs are seen in the lowest quintile of energy intake, which includes mostly vegetables 130 and some fruits as shown in Supplemental Table 1” => the term “energy intake” refers to a diet! Did you mean the lowest quintile of food group’s energy density?
Line 132: remove the word processed and ultra-processed from the result, as you didn’t explain in the methods how you decide that that food is processed or ultra-processed (which algorithm?).
Line 135: energy cost (not cost)
Line 136: energy cost (not cost)
Line 137: energy cost (not cost)
Figure 1: y axis : is it in dollars or in Mexican pesos? Figure 1 title: indicate that quintiles were calculated on n=222 food categories (if it is the case, still not clear for me)
Figure 2 title is not understandable: 10000 kcal of what? Monthly income of whom?
Affordability index is rather an in-affordability index (the higher it is, the less affordable is the food). Could you give an example of how it could be interpreted: is it the percentage of income needed to purchase 10000 kcal of food? But what kind of food? Calculated how? If it is calculated for household’s purchases, does this means that households in the lowest income quintile would need more than 10% of their income to by 10000kcal of (overall) foods???
If it is the case you are comparing the purchase of 10000 kcal of differing nutritional quality, as it is likely that 10000kcal purchased by households from the lowest income quintile have lower nutritional quality than 10000kcal purchased by households from other income quintiles
Line 181: rephrase the first sentence of the discussion (energy cost not cost; energy density as not calculated for beverages)
Lines 143-145: The affordability index by quintile of energy density shows that foods in the high and highest quintiles of energy density PROVIDE more affordable kcal: less income is needed to purchase KCAL of foods with a higher energy density.
Lines 145-146: In contrast, as expected, foods with the lowest energy density (vegetables and some fruits) are less affordable KCAL
Lines 151-157: “Descriptive analyses by income quintile show that in 2016, lower income households had to spend 29.5% of their monthly income to purchase 10,000 kcal of foods from the lowest energy dense quintile, compared to 4% for the highest income group (Supplemental Tables S3 and S4). In contrast, etc.”
=> In table S3 I read 28.6% and not 29.5%
=> Those results are clear and easily understandable. I strongly recommend to exchange results with the affordability index which is difficult to understand (in particular because it is a non-affordability index) with results on the percentage of income to buy 10000 kcal of foods in the main text. This would imply to put tables S3 and S4 in the main text (possibly transformed in figures) and to transfer Figures 1 and 2 into supplemental materials.
Or perhaps affordability = percent of income to buy 10000 kcal (as stated in the title of Table1). But if is the case, why do you use two different terminologies for the same thing? Unclear.
Table 1 seems to show overall affordability of food, whatever the nutritional quality. It would seem more logical to start with this result and then to show that this is mainly driven by the decreasing affordability of food with high nutritional quality and low energy density. The paper would be much more easy to follow.
Line 161: lowest INCOME quintile.
Table 1 (and line162): 10000 Kcal of food and beverages calculated how?????????????? Are the 10000kcal comparable across income quintiles?
Line 181: energy cost (not cost)
Line 184: energy cost (not prices)
Line 186-187: as said above, it would be much more logical to start with this result and then explain it by the differential evolution of food affordability according to nutritional quality
Line 217: CALORIES FROM food and beverages
Line 222-225: it seems clear that affordability of foods has decreased mainly because affordability of healthy food has decreased. But this would be more robust if a non-relative nutrient profiling system had been used to classify foods in healthy and less healthy…
Line 234-236: regarding free sugars, it would be worth mentioning that some nutrient profiling systems now use added sugars in their algorithm (eg the SENS algorithm, Eur J Clin Nutr, 2017; 72, 236–248.)
Lines 244-245: Not so easy. Beware that taxes (and un-focused subsidies) may increase socioeconomic inequalities (see Int J Behav Nutr Phys Activity, 2014;11:66.)
Line 248: is it safe to drink tap water in Mexico, especially in poor neighborhoods?
Line 254: You sentence is about intervention but ref 42 is not an intervention study. In-store intervention studies conducted in real settings with positive influence on food purchasing behaviors are scarce. I know at least one successful intervention study conducted in discount stores which combined shelf labeling with a social marketing strategy to promote inexpensive foods with good nutritional quality (Gamburzew A, Darcel N, Gazan R, Dubois C, Maillot M, Tomé D, Raffin S, Darmon N. In-store marketing of inexpensive foods with good nutritional quality in disadvantaged neighborhoods: increased awareness, understanding and purchasing. Int J Behav Nutr Phys Activity, 2016, 13:104.), and I there are also very inspiring in-store interventions in poor populations conducted by John Gittelsohn, and colleagues.
Reviewer 2 Report
This manuscript aimed to assess cost and affordability of food in Mexico as a function of energy density and nutrient quality over time. More energy dense foods and food with lower nutrient quality cost less. Lower energy density foods and high nutrient foods became less affordable over time. The work also looked at the affordability of buying 10,000kcal of food. This is an interesting study that addresses an important question in a novel setting (Mexico), however a number of aspects of the paper need revising.
The main focus of the paper is unclear. There is heavy emphasis throughout the paper on looking at cost and affordability in energy dense foods and high nutrient foods, yet these results (figures S1 and S2) are in the supplementary materials and no statistical analysis has been done. The paper conclusions draw heavily on these figures despite no statistical analysis. In contrast, examining the affordability of purchasing 10,000kcal relative to income is given much less focus in the manuscript (e.g. introduction, abstract, discussion), yet the graphs are presented in the main manuscript and regression analyses are conducted (table 1, figure 2).
Further, Tables S3 and S4 in the supplementary materials are not referred to in the manuscript or discussed in any way. These should be removed or utilised in the manuscript.
Basic information about the study sample is missing – how big was the sample, what were their demographics?
Table 1 needs to be clearer. The values should be clearly labelled as the regression coefficients (standardised or unstandardised beta), and confidence intervals should be given.
There is no data availability statement with this manuscript.
The study mentioned in the introduction on lines 47-50 (reference number 15) appears to partially contradict expectations i.e. higher cost diets contained less quality food (energy dense, processed food and sugar sweetened beverages). It would be good to discuss this further in the context of this work.
The authors mention that their analyses do not include food consumed away from home, but it would be good to discuss how this might affect interpretation of the study’s results i.e. should they be considered more cautiously because a key aspect of people’s diet isn’t included, and how might including that data affect the pattern of results?
It is not clear to me why the authors aggregated data into quintiles. Keeping the data as continuous would maintain greater statistical power.
On line 133 there is a claim of significant effect when no statistical testing has been done.
Line 146 says ‘as expected’ but the authors did not make any predictions to begin with.
On lines 214 to 217, the discussion describes research showing that high energy dense food costs less. These studies should also be described in the introduction of the manuscript so that the reader can understand how these informed the current work. This would also help the reader understand the originality of the work and what this work contributes to the scientific literature.
Lines 223-226 – Is this something that can be looked at in this study, since you have income data across time?
Line 199 ‘most used’ needs a reference.
Line 147, should this be ‘lowest quintile’?
Round 2
Reviewer 1 Report
Line 117: « not purchased by a household »=> I don’t understand: why « not purchased” and which “household” => Do you mean “purchased by at least one household” ???
My comment “The interest of calculating quintiles of NRFI is debatable. One would rather know which foods have a good nutritional quality in an absolute sense, and not which foods have a better nutritional quality than others” was not correctly addressed in the paragraph 267-272. I never asked to analyse data food category by food category. I proposed to use a nutrient profiling system able to categorize food in an absolute way, so that it would not be sensitive to trends in the nutritional quality of the food offer.
My concern with relative vs absolute nutrient profile assessment is important because you are analyzing trends over time. You didn’t explain how you calculated quintiles, but I assume that you recalculated them at each point of time. Therefore, if the nutrient profile of the food is decreasing or increasing over time, you won’t be able to know whether your results on affordability are due to changes in the nutrient profile of foods or to changes in food prices over time. It is the reason why I was insisting in using a system not defining nutritional quality in a relative way. Alternatively, you could calculate quintiles in 1994 and then use the same 1994 NRFI or ED quintile values for the other years (i.e. at each round, the same quintile values), in order to compare what is comparable, i.e. the evolution of the affordability of foods having strictly similar nutrient profile. In this case, you should also indicate somewhere how quintiles values have changed with time.
Reviewer 2 Report
Thank you for making changes to the manuscript, I have just a few more minor requests to improve the clarity of the paper.
Line 137 says ‘described by Zhen’, is this intended to be a citation?
Please add to all figures and tables that show the affordability index that higher scores reflect lower affordability.
Please add sample sizes/n values to Table S3 descriptives of the sample.
Further, the sample size information on lines 168-173 should be moved to be the first paragraph in the results section.
You should refer to the statistical results for NRFI and energy density in Table 1 in the relevant section of the text (section 3.1 affordability index by quintile of energy density and quintile of NRFI).
Line 205 – please make this clearer that an increase in the affordability index means that items are less affordable. Line 206 says ‘increase in affordability’ but I think this should be either ‘increase in affordability index’ or ‘decrease in affordability’.
The sentence on line 217-219 doesn’t make sense/isn’t finished.
Round 3
Reviewer 1 Report
Here it is
I am sorry but I still don't understand the sentence Therefore, to estimate the affordability index, given that households do not buy all products, we imputed within each municipality the energy costs of food and beverages not purchased by a household during the week of data collection.
in authors answers below:
Line 117: « not purchased by a household »=> I don’t understand: why « not purchased” and which “household” => Do you mean “purchased by at least one household” ???
Response: The unit of analysis in the ENIGH is the household. Households have to report all food and beverages purchased daily for a week. During that week, they purchase food and beverages but they do not buy all the products in the market. We added a sentence in the data source section and modified the paragraph mentioned by the reviewer as follows:
“The affordability index illustrates how much a food or beverage cost per 1,000 kcal, regardless of whether the household purchased that food or beverage. Therefore, to estimate the affordability index, given that households do not buy all products, we imputed within each municipality the energy costs of food and beverages not purchased by a household during the week of data collection. The higher the affordability index is, the less affordable a food or beverage is.”
Reviewer 2 Report
Thank you for making these changes, I have no further requests.
